# Machine Learning Dynamic Ensemble Methods for Solar Irradiance and Wind Speed Predictions

**Francisco Diego Vidal Bezerra [1], Felipe Pinto Marinho [2], Paulo Alexandre Costa Rocha [1,3,*], Victor Oliveira Santos [3], Jesse Van Griensven Thé [3,4] and Bahram Gharabaghi [3]**

[1] Department of Mechanical Engineering, Technology Center, Federal University of Ceará, Fortaleza 60020-181, CE, Brazil; diegobez@alu.ufc.br

[2] Department of Teleinformatics Engineering, Technology Center, Federal University of Ceará, Fortaleza 60020-181, CE, Brazil; fpmarinho@alu.ufc.br

[3] School of Engineering, University of Guelph, 50 Stone Rd. E, Guelph, ON N1G 2W1, Canada; volive04@uoguelph.ca (V.O.S.); jesse.the@weblakes.com (J.V.G.T.); bgharaba@uoguelph.ca (B.G.)

[4] Lakes Environmental, 170 Columbia St. W, Waterloo, ON N2L 3L3, Canada

[*] Correspondence: paulo.rocha@ufc.br

**Abstract:** This paper proposes to analyze the performance increase in the forecasting of solar irradiance and wind speed by implementing a dynamic ensemble architecture for intra-hour horizon ranging from 10 to 60 min for a 10 min time step data. Global horizontal irradiance (GHI) and wind speed were computed using four standalone forecasting models (random forest, k-nearest neighbors, support vector regression, and elastic net) to compare their performance against two dynamic ensemble methods, windowing and arbitrating. The standalone models and the dynamic ensemble methods were evaluated using the error metrics RMSE, MAE, $R^2$, and MAPE. This work's findings showcased that the windowing dynamic ensemble method was the best-performing architecture when compared to the other evaluated models. For both cases of wind speed and solar irradiance forecasting, the ensemble windowing model reached the best error values in terms of RMSE for all the assessed forecasting horizons. Using this approach, the wind speed forecasting gain was 0.56% when compared with the second-best forecasting model, whereas the gain for GHI prediction was 1.96%, considering the RMSE metric. The development of an ensemble model able to provide accurate and precise estimations can be implemented in real-time forecasting applications, helping the evaluation of wind and solar farm operation.

**Keywords:** wind energy; solar energy; renewable energy; machine learning; forecasting ensembles

## 1. Introduction

Electricity generated from fossil fuel sources has been the main driver of climate change, probably contributing over 70% of greenhouse gas emissions and over 90% of all carbon gas emissions. The alternative of decarbonizing the world's electricity generation system is focused on being alert to sources of renewable energy, whose generation costs are increasingly accessible [1].

The impact of intermittency generation [2] on the electrical grid is an undesired effect when it comes to electrical generation from alternative energy resources, such as wind speed and global solar radiation. Since this generation is dependent on weather conditions, one of the means to eliminate or reduce its uncertainties is the availability of good prediction methods for these resources [3].

The search for parameters that can describe atmospheric behavior and its predictability has led research on machine learning to develop and to create models, based on the most diverse types of predictors, for use in different areas. In [4], multilayer machine learning is used to improve the resolution of ground-based astronomical telescopes. In [5], parameters are used to construct an atmospheric circulation model.

The influences of atmospheric factors on the generation of electrical energy from solar and wind sources are usually the main problem in the generation of smart grids, where large-scale generation plants need to be integrated into the electrical grid, which directly affects the planning, investment, and decision-making processes. Forecast models can minimize that problem via machine learning models [6].

The benefits of optimizing the forecast of generation from wind and solar sources using models is also an economic factor, as it gives greater security to the electricity sector via the improvement of renewable energy purchase contracts [7].

A 14-year-long data set was explored in [8], containing daily values of meteorological variables. This dataset was used to train three deep neural network (DNNs) architectures over several time horizons to predict global solar radiation for Fortaleza, in the northeastern region of Brazil. The accuracy of the predictions was considered excellent according to its normalized root mean squared error (nRMSE) values and good according to mean absolute percentage error (MAPE) values.

The variability of mathematical prediction models has individual importance inherent to each one of the methods employed, and in this scenario, dynamic ensemble models emerge, which present potentially better performance when compared to individual models, since they seek maximum optimization by considering the best of the individual models. This approach is currently very successfully used in research and industrial areas. Several dynamic ensemble methods have been developed for forecasting energy generation from renewable sources in which they use the presence of well-known forecast models such as random forest regression (RF), support vector regression (SVR), and k-nearest neighbors (kNN), which are applied to integrate optimizations for use in dynamic ensemble methods [9].

The random forest (RF) forecasting model is based on the creation of random decision trees. In this method, these decision trees state specific rules and conditions for the flow of the result until its conclusion.

The support vector machine (SVR) is a regression algorithm that uses coordinates for individual observations and uses hyperplanes to segregate data sets. This is a widely used method for categorizing clusters and classifying. This model was first developed for classification purposes and has been largely tested [10,11] in recent approaches [12] to develop a novel method for the maximum power point tracking of a photovoltaic panel and in [13], where solar radiation estimation via five different machine learning approaches is discussed.

The KNN method is a supervised learning algorithm which is widely used as a classifier that, based on the proximity of nearest neighboring data, performs categorization via similarity and predicts a new sample using the K-closest samples. Recently, this approach has been used in [14], where virtual meteorological masts use calibrated numerical data to provide precise wind estimates during all phases of a wind energy project to reproduce optimal site-specific environmental conditions.

Most studies have focused on accurate wind power forecasting, where the random fluctuations and uncertainties involved are considered. The study in [15] proposes a novel method of ultra-short-term probabilistic wind power forecasting using an error correction modeling with the random forest approach.

The elastic net method is a regularized regression method that linearly combines the penalties of the LASSO and Ridge methods. In [16], the study uses forecast combinations that are obtained by applying regional data from Germany for both solar photovoltaic and wind via the elastic net model, with cross-validation and rolling window estimation, in the context of renewable energy forecasts.

The state of the art is currently to use dynamic ensemble methods in a meta-learning approach such as arbitrating, which uses output combinations according to the predictions of the loss that shall result, as well as windowing approaches, which have parameterizations for adjusting the degree of data to be considered [17].

In [18], a global climate model (GCM) is studied to improve a near-surface wind speed (WS) simulation via 28 coupled model intercomparisons using dynamical components.

In [19], a hybrid transfer learning model based on a convolutional neural network and a gated recurrent neural network is proposed to predict short-term canyon wind speed with fewer observation data. The method uses a time sliding window to extract time series from historical wind speed data and temperature data of adjacent cities as the input of the neural network.

In [20], authors studied the multi-GRU-RCN method, an ensemble model, to obtain significant information regarding factors such as precipitation and solar irradiation via short-time cloud motion predictions from a cloud image. The ensemble modeling used in [21] integrates wind and solar forecasting methodologies applied to two locations at different latitudes and with climatic profiles. The obtained results reduce the forecast errors and can be useful in optimizing planning to use intermittent solar and wind resources in electrical matrices.

A proposed new ensemble model in [22] was based on graph attention networks (GAT) and GraphSAGE to predict wind speed in a bi-dimensional approach using a Dutch dataset including several time horizons, time lags, and weather influences. The results showed that the ensemble model proposed was equivalent to or outperformed all benchmarking models and had smaller error values than those found in reference literature.

In [23], time horizons ranging from 5 min to 30 min were studied in 5-min time steps in evaluating solar irradiance short-term forecasts to global horizontal irradiance (GHI) and direct normal irradiance (DNI) using deep neural networks with 1-dimensional convolutional neural networks (CNN-1Ds), long short-term memory (LSTM), and CNN–LSTM. The metrics used were the mean absolute error (MAE), mean bias error (MBE), root mean squared error (RMSE), relative root mean squared error (rRMSE), and coefficient of determination ($R^2$). The best accuracy was obtained for a horizon of 10 min, improving 11.15% on this error metric compared to the persistence model.

There are studies employing different DNN architectures, such as GNN, CNN, and LSTM, achieving satisfactory outcomes in different fields of science [24–27]. However, the present work focuses on classical ML, since the main objective is to identify the best supporting ensemble approach to the ML procedures. by analyzing the influence of dynamic ensemble arbitrating and windowing methods on machine learning algorithms traditionally, focusing on predicting electrical power generation. We also present their greater efficiency, using data of interest for energy production with input variables of wind speed and solar irradiance. We have followed this approach because of its advantage in exploring dynamic ensemble methods, since these seek the best pre-existing efficiency for generating a unique and more effective predictability model.

## 2. Location and Data

In this paper, two data types were used to carry out the analysis, which were acquired from solarimetric and anemometric stations located in Petrolina—PE. The data were collected from the SONDA network (National Organization of Environmental Data System) [28], which was a joint collaboration between several institutions and was created for the implementation of physical infrastructure and human resources, aimed at raising and improving the database of solar and wind energy resources in Brazil.

The time sampling used in this study was 10 min, and the duration of data collection was from January 2007 to December 2010. The detailed information about the data of the solarimetric and anemometric station is shown in Table 1, where MI (min) is the "measurement interval" and the duration of data collection is presented as MP, "measured period". Its location on the map is shown in Figure 1.

**Table 1.** Geographic coordinates, altitude in relation to the sea level, measurement intervals, and measurement periods of the data were collected from the Petrolina station. MI and MP stand for, respectively, "measurement interval" and "measurement period".

| Type | Lat. (○) | Long. (○) | Alt. (m) | MI (min) | MP |
|---|---|---|---|---|---|
| Anemometric | 09°04′08″ S | 40°19′11″ O | 387 | 10 | 1 January 2007 to 12 December 2010 |
| Solarimetric | | | | | 1 January 2010 to 12 December 2010 |

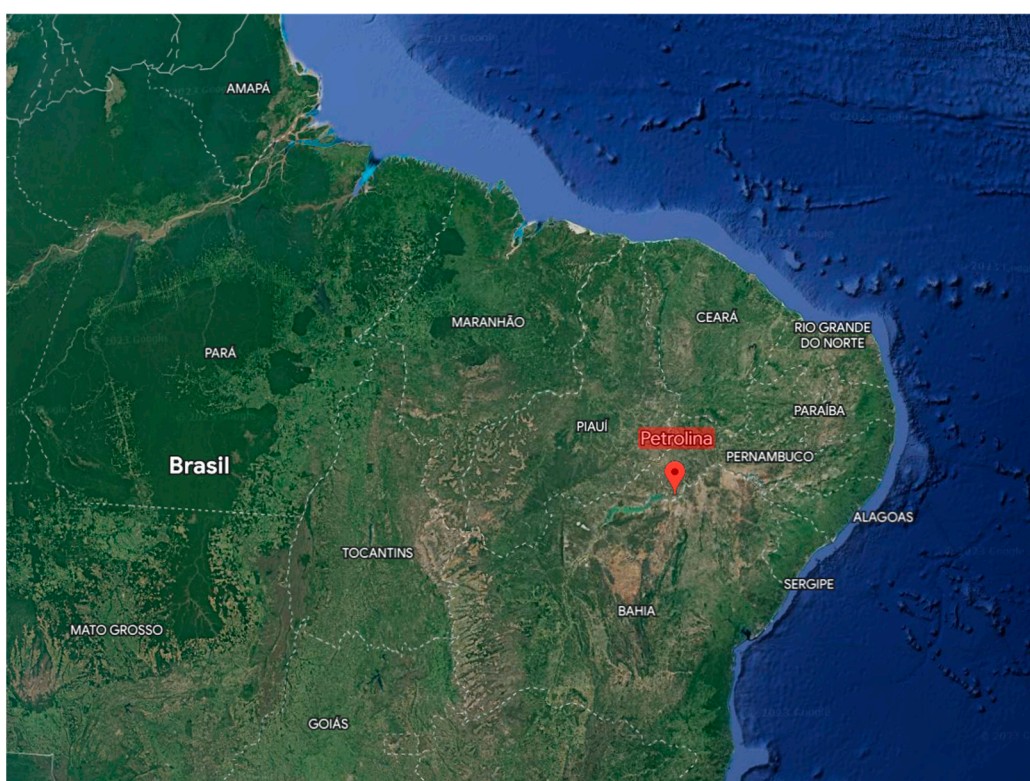

**Figure 1.** Map of the northeast of Brazil. The Petrolina measurement site is highlighted [29].

The Petrolina region is classified as a BSh Koppen climate zone [30]. There are considerable differences in the annual cycle between solar radiation and wind. The average wind speed and solar irradiance in Petrolina experience significant seasonal variations throughout their annual cycle. The windiest interval of the year occurs from May to November, with average wind speeds above 5.4 m/s. The month with the strongest winds is August, with an average hourly wind speed of 6.7 m/s. The period with the lowest wind volume of the year is from November to May. The month with the calmest winds is March, with an average hourly wind speed of 4.1 m/s.

The period of greatest solar radiance in the year is from September to November, with a daily average above 7.2 kWh/m$^2$, with October being the peak with an average of 7.5 kWh/m$^2$. The period with the lowest solar radiance in the year is from May to July, with a daily average of 6.1 kWh/m$^2$, with June being the month with the lowest solar radiance, with an average of 5.7 kWh/m$^2$.

### 2.1. Wind Speed Data

The wind speed were was obtained in m/s from a meteorological station, which has anemometric sensors at altitudes of 25 m and 50 m from the ground. The highest altitude was chosen for this study, both to reduce the effects of the terrain and to be closer to the altitudes currently in practice for wind turbines [31].

*2.2. Irradiance Data*

The global horizontal irradiance (GHI) data acquired from the solarimetric station were used in this study, and the clear-sky coefficient was considered, in order to remove dependence on air mass in the irradiance values that reach the sensors [32], through the use of the clear-sky factor ($I_{cs}$) [33], using the polynomial fit model [34]. The work [35] obtained promising results from the same database using two machine learning estimation models for (GHI).

In order to obtain irradiance data independent of air mass variations, we used $k_t$, which is defined by the ratio between the global horizontal irradiance value (GHI) (I) and the clear sky factor ($I_{cs}$), as shown in Equation (1).

$$k_t = \frac{I}{I_{cs}} \tag{1}$$

**3. Methodology**

Initially, wind speed and irradiance data were acquired and the intervals for the test and training sets were determined. For wind speed data, in a measurement period from 2007 to 2010, the first three years were used as the training data set and the last year as the test set. In order to allow the evaluation of the performance of the tested forecasting models and also of dynamic ensemble methods, this study developed a computational code in Python to evaluate the output values obtained by the well-known machine learning forecasting methods: random forest, k-nearest neighbors (kNN), support vector regression (SVR), and elastic net. For each of the methods, the best performance parameters (lower root mean squared error (RMSE)) were evaluated. Right after the stage of acquisition and determination of the optimal parameters for each of the models, the methods of dynamic ensemble windowing and arbitrating were executed, from which performance metrics values were also obtained: coefficient of determination ($R^2$), root mean squared error (RMSE), mean absolute error (MAE) and mean absolute percentage error (MAPE). These values were compared to evaluate the efficiency of the dynamic ensemble methods compared to other stand-alone models. The variation of the λ parameter for windowing, which is the length used for the extension of the values considered in the data forecast, was also evaluated. The methodology used can be seen in Figure 2.

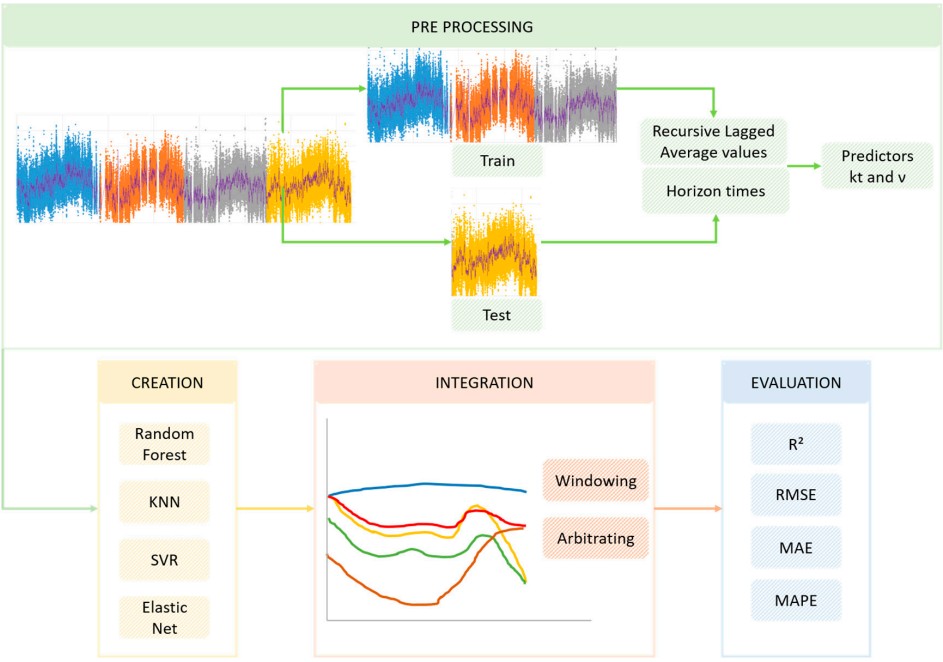

**Figure 2.** Diagram of the data flow for the applied methodology.

In the data pre-processing, a recursive approach of Lagged Average values for $k_t$ and $v$ time series was applied: this feature is given by the vector L(t) with components calculated using Equation (2).

$$L_i(t) = \frac{1}{N} \sum_{t \in [t-i\delta-T, t-(i-1)\delta-T]} x(t) \tag{2}$$

### 3.1. Windowing Method

The diversity of the models makes the forecast analysis rich and complex, since each model has strong points and other weaknesses, in the sense that from this combination, the best results can be treated and considered to obtain more accurate forecasts. To perform this combination, it is necessary to know how to estimate at which points certain specific models perform better.

Windowing [17] is a dynamic ensemble model, where weights are calculated based on the performance of each individual model, evaluated in a data window referring to immediately previous data. The size of this window is parameterized by the λ value. This means that the weights of each model are re-evaluated at each time step, and then they are classified to catalogue only the best performance results, generating a hybrid model.

### 3.2. Arbitrating Method

Arbitrating [36] uses the metalearning method to learn and predict the classifiers. In this study, it regards the weights based on each model's performance for a given time step. At each simulation instant, the most reliable model is selected and included in the prediction process.

### 3.3. Machine Learning Prediction Models and Dynamic Ensemble Method Parameters

In the data training stage, GridSearch was used with 5-fold cross-validation. The search parameters are shown in Table 2.

**Table 2.** Search parameters and grid values applied to the tested methods.

| Method | Search Parameter | Grid Values |
|---|---|---|
| Random forest | maxdepth | [2, 5, 7, 9, 11, 13, 15, 21, 35] |
| KNN | nearest neighbours k | $1 \leq k \leq 50$, k integer |
| SVR | penalty term C | [0.1, 1, 10, 100, 1000] |
| | coefficient λ | [1, 0.1. 0.01, 0.001, 0.0001] |
| Elastic net | regularization term λ | [1, 0.1. 0.01, 0.001, 0.0001] |
| Windowing | Λ | [1, 3, 6, 12, 25, 50, 100] |
| Arbitrating | | * |

*: Due to the use of a meta-heuristic methodology, the initial parameter was not needed.

GridSearch is a tool from the Scikit-learn library used in Python which applies a methodology whose function is to combine parameters from the methods under evaluation and present them in a single output object for analysis. This is a very important tool when comparing performance between methods, the object of this study.

### 3.4. Performance Metrics Comparison Criteria

As the purpose of this work is to evaluate the performance of dynamic ensemble methods against other methods, performance metrics had to be determined to allow it. The metrics used were those of Equations (3)–(6).

- Coefficient of determination ($R^2$)

$$R^2 = 1 - \frac{\sum_{i=1}^{N} (y_i - \hat{y_l})^2}{\sum_{i=1}^{N} (y_i - \overline{y_l})^2} \tag{3}$$

- Root mean squared error (*RMSE*)

$$RMSE = \sqrt{\sum_{i=1}^{N} \frac{(y_i - \hat{y_l})^2}{n}} \tag{4}$$

- Mean absolute error (*MAE*)

$$MAE = \frac{1}{N} \sum_{i=1}^{N} |\hat{y_l} - y_i| \tag{5}$$

- Mean absolute percentage error (*MAPE*)

$$MAPE = \frac{1}{n} \sum_{i=1}^{n} \left| \frac{y_i - \hat{y_l}}{y_i} \right| \tag{6}$$

## 4. Results and Discussion

This section discusses the results generated in this work. It focuses on the analysis of efficiency metrics for the machine learning methods employed. This analysis determines which method/parameters obtain the best performance in the application of wind speed and solar irradiance data.

### 4.1. Wind Speed Predictions

During the search for best-performance methods, the optimized parameters for each of the tested methods needed to be identified. This allowed for the elaboration of the dynamic ensemble, which was built upon the merging of the best-performance results at each time step and for all the methods in question. The optimal parameters found for each of the time horizons are shown in Table 3.

**Table 3.** Best parameters for each machine learning method.

| Method | Parameter | t + 10 | t + 20 | t + 30 | t + 60 |
|---|---|---|---|---|---|
| Random forest | best_max_depth | 7 | 7 | 7 | 7 |
| | best_n_estimators | 20 | 20 | 20 | 20 |
| KNN | best_n_neighbors | 49 | 49 | 49 | 49 |
| SVR | best_C | 1 | 1 | 1 | 1 |
| | best_epsilon | 0.1 | 0.1 | 1 | 0.1 |
| Elastic net | best_l1_ratio | 1 | 1 | 1 | 1 |

Efficiency evaluations for each of the forecasting methods were based on performance metrics evaluations for each time horizon under study (t + 10, t + 20, t + 30 and t + 60). Initially, for all time horizons, windowing proved to be the most efficient method. Then, a fine-tuning evaluation was performed based on the variation of the windowing parameter to assess its influence on performance. The predominance of better performance for windowing in all time horizons and its comparisons can be seen in Table 4 and Figure 3.

**Table 4.** Comparison of RMSE (m/s) values, using different methods for different time horizons and windowing $\lambda$ parameter variation. The best results for each time horizon are in bold.

| Time Horizon | $\lambda$ | RF | KNN | SVR | Elastic Net | Windowing | Arbitrating |
|---|---|---|---|---|---|---|---|
| | 1 | | | | | 0.69263 | |
| | 3 | | | | | 0.69180 | |
| | 6 | | | | | 0.69114 | |
| | 12 | | | | | 0.69041 | |
| t + 10 min | **19** | 0.69458 | 0.71040 | 0.69396 | 0.69828 | **0.69007** | 0.69447 |
| | 25 | | | | | 0.69040 | |
| | 50 | | | | | 0.69226 | |
| | 74 | | | | | 0.69402 | |
| | 100 | | | | | 0.69431 | |
| | 1 | | | | | **0.86817** | |
| | 3 | | | | | 0.87353 | |
| | 6 | | | | | 0.87563 | |
| t + 20 min | 12 | 0.88310 | 0.89332 | 0.88372 | 0.88554 | 0.87699 | 0.88315 |
| | 25 | | | | | 0.87803 | |
| | 50 | | | | | 0.87889 | |
| | 100 | | | | | 0.87960 | |
| | 1 | | | | | **0.97497** | |
| | 3 | | | | | 0.98017 | |
| | 6 | | | | | 0.98333 | |
| t + 30 min | 12 | 0.99469 | 0.99859 | 0.99130 | 0.99660 | 0.98583 | 0.99091 |
| | 25 | | | | | 0.98702 | |
| | 50 | | | | | 0.98832 | |
| | 100 | | | | | 0.98902 | |
| | 1 | | | | | **1.15150** | |
| | 3 | | | | | 1.15647 | |
| | 6 | | | | | 1.16170 | |
| t + 60 min | 12 | 1.18092 | 1.19527 | 1.17764 | 1.18281 | 1.16685 | 1.18156 |
| | 25 | | | | | 1.16987 | |
| | 50 | | | | | 1.17254 | |
| | 100 | | | | | 1.17455 | |

Elastic net is a penalized linear regression model that is a combination of Ridge and LASSO regression into a single algorithm and uses best_l1_ratio as a penalty parameter during the training step, being 0 for Ridge and 1 value for LASSO regression. From Table 3, the parameter obtained the value of 1, which means that LASSO regression was used in its entirety.

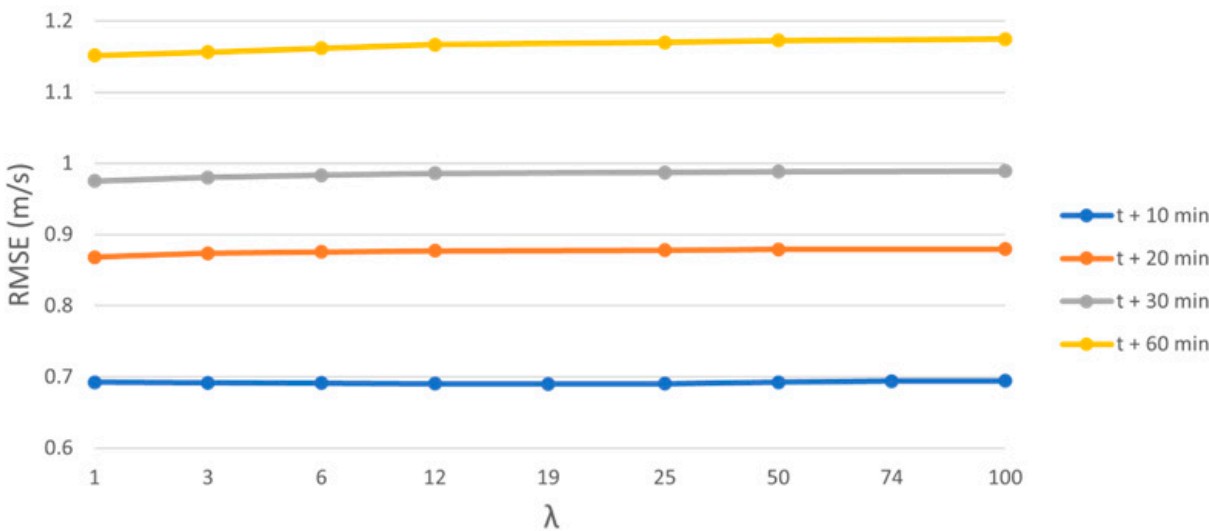

**Figure 3.** Windowing λ parameter variation influence in RMSE for different time horizons in wind speed data analysis for all the studied time horizons.

As with the evaluation employing RMSE, values from $R^2$, MAE, and MAPE were also assessed. Once the best performance was found for the windowing ensemble method, an in-depth analysis was performed based on the variation of its parameter λ to assess the influence on its internal performance. Since the time horizon that presented the best performance was t + 10, this was the focus of the analysis, as shown in Figures 4–7. The detailed data for all the horizons is shown in Tables 5–7.

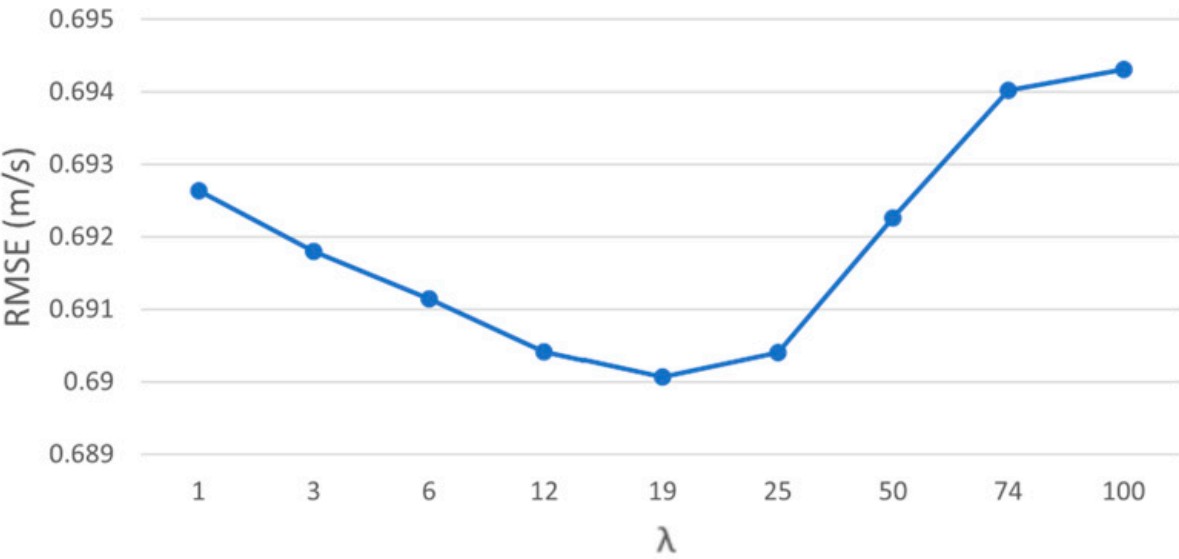

**Figure 4.** Windowing λ parameter influence on RMSE value for the time horizon t + 10.

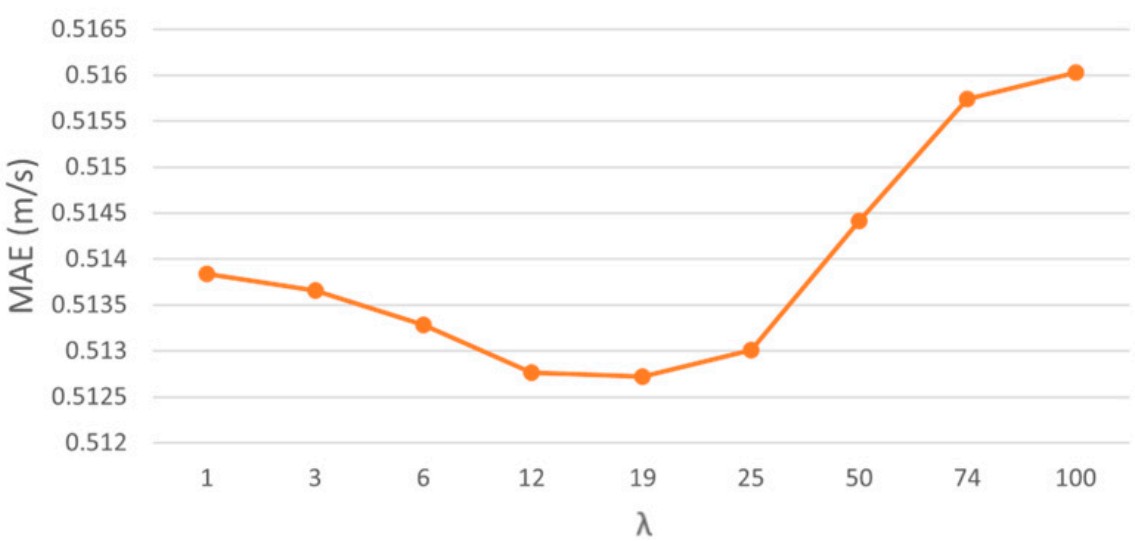

**Figure 5.** Windowing λ parameter influence in MAE value for the time horizon t + 10.

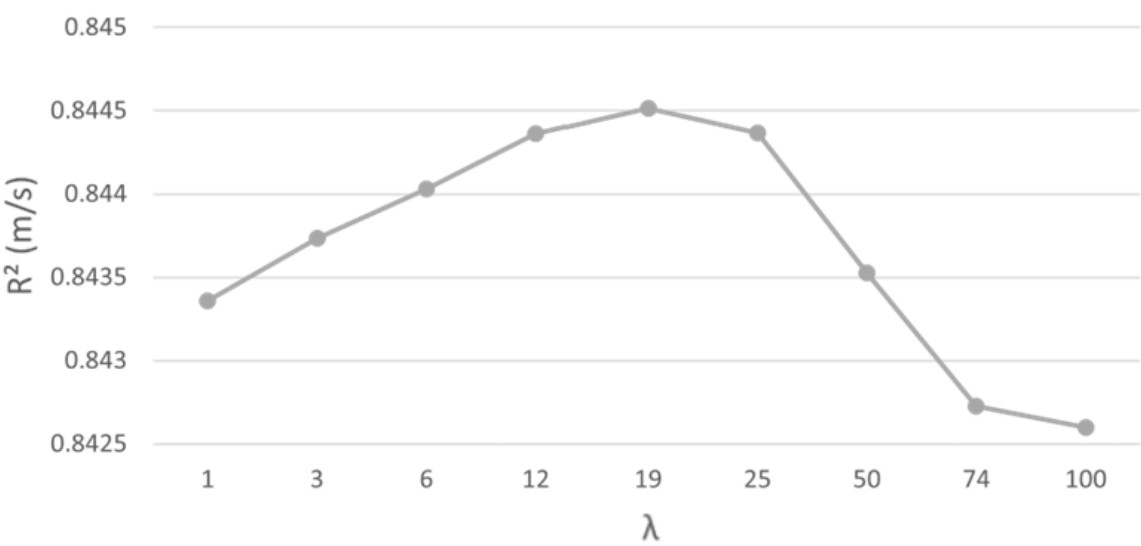

**Figure 6.** Windowing λ parameter influence in $R^2$ value for the time horizon t + 10.

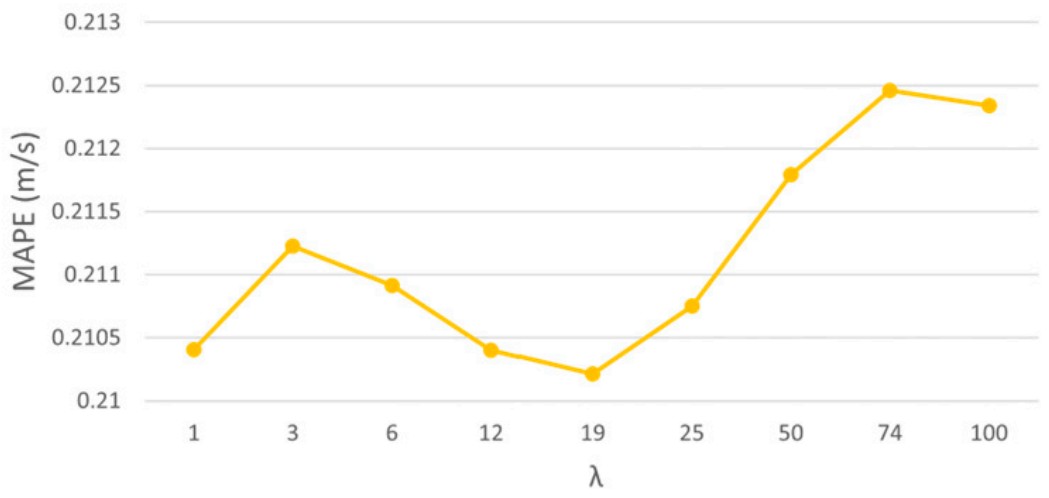

**Figure 7.** Windowing λ parameter influence in MAPE value for the time horizon t + 10.

**Table 5.** Comparison of MAE (m/s) values, using different methods in different time horizons and windowing λ parameter variation. The best results for each time horizon are in bold.

| Time Horizon | λ | RF | KNN | SVR | Elastic Net | Windowing | Arbitrating |
|---|---|---|---|---|---|---|---|
| t + 10 min | 1 | | | | | 0.51384 | |
| | 3 | | | | | 0.51366 | |
| | 6 | | | | | 0.51328 | |
| | 12 | | | | | 0.51276 | |
| | **19** | 0.51592 | 0.53216 | 0.51438 | 0.51853 | **0.51272** | 0.51711 |
| | 25 | | | | | 0.51301 | |
| | 50 | | | | | 0.51441 | |
| | 74 | | | | | 0.51574 | |
| | 100 | | | | | 0.51603 | |
| t + 20 min | 1 | | | | | **0.64663** | |
| | 3 | | | | | 0.65140 | |
| | 6 | | | | | 0.65332 | |
| | 12 | 0.65845 | 0.66882 | 0.66040 | 0.65990 | 0.65435 | 0.65936 |
| | 25 | | | | | 0.65554 | |
| | 50 | | | | | 0.65637 | |
| | 100 | | | | | 0.65695 | |
| t + 30 min | 1 | | | | | **0.72594** | |
| | 3 | | | | | 0.73105 | |
| | 6 | | | | | 0.73402 | |
| | 12 | 0.74250 | 0.74735 | 0.74125 | 0.74347 | 0.73625 | 0.74097 |
| | 25 | | | | | 0.73732 | |
| | 50 | | | | | 0.73846 | |
| | 100 | | | | | 0.73902 | |
| t + 60 min | 1 | | | | | **0.86784** | |
| | 3 | | | | | 0.87277 | |
| | 6 | | | | | 0.87826 | |
| | 12 | 0.89496 | 0.90753 | 0.89179 | 0.89589 | 0.88307 | 0.89570 |
| | 25 | | | | | 0.88580 | |
| | 50 | | | | | 0.88813 | |
| | 100 | | | | | 0.88963 | |

When we checked the influence of the λ parameter on windowing method performance, it was found from λ = 74 that it is no longer the most efficient method, and SVR becomes the best one, due to its lowest RMSE value. It is important to highlight that the best performance value for the windowing method, which is the best performance overall, was found for λ = 19. The performance comparison between the two methods can be seen in Figure 8.

**Table 6.** Comparison of $R^2$ (m/s) values, using different methods in different time horizons and windowing λ parameter variation. The best results for each time horizon are in bold.

| Time Horizon | λ | RF | KNN | SVR | Elastic Net | Windowing | Arbitrating |
|---|---|---|---|---|---|---|---|
| | 1 | | | | | 0.84336 | |
| | 3 | | | | | 0.84373 | |
| | 6 | | | | | 0.84403 | |
| | 12 | | | | | 0.84436 | |
| t + 10 min | **19** | 0.84248 | 0.83522 | 0.84275 | 0.84079 | **0.84451** | 0.84252 |
| | 25 | | | | | 0.84436 | |
| | 50 | | | | | 0.84353 | |
| | 74 | | | | | 0.84273 | |
| | 100 | | | | | 0.84260 | |
| | 1 | | | | | **0.75388** | |
| | 3 | | | | | 0.75083 | |
| | 6 | | | | | 0.74963 | |
| t + 20 min | 12 | 0.74534 | 0.73941 | 0.74498 | 0.74393 | 0.74885 | 0.74531 |
| | 25 | | | | | 0.74825 | |
| | 50 | | | | | 0.74776 | |
| | 100 | | | | | 0.74736 | |
| | 1 | | | | | **0.68958** | |
| | 3 | | | | | 0.68626 | |
| | 6 | | | | | 0.68423 | |
| t + 30 min | 12 | 0.67690 | 0.67436 | 0.67909 | 0.67566 | 0.68262 | 0.67935 |
| | 25 | | | | | 0.68186 | |
| | 50 | | | | | 0.68102 | |
| | 100 | | | | | 0.68057 | |
| | 1 | | | | | **0.56685** | |
| | 3 | | | | | 0.56310 | |
| | 6 | | | | | 0.55914 | |
| t + 60 min | 12 | 0.54443 | 0.53329 | 0.54695 | 0.54297 | 0.55522 | 0.54393 |
| | 25 | | | | | 0.55291 | |
| | 50 | | | | | 0.55087 | |
| | 100 | | | | | 0.54933 | |

**Table 7.** Comparison of MAPE (m/s) values, using different methods in different time horizons and windowing λ parameter variation. The best results for each time horizon are in bold.

| Time Horizon | λ | RF | KNN | SVR | Elastic Net | Windowing | Arbitrating |
|---|---|---|---|---|---|---|---|
| | 1 | | | | | 0.21040 | |
| | 3 | | | | | 0.21122 | |
| | 6 | | | | | 0.21092 | |
| | 12 | | | | | 0.21040 | |
| t + 10 min | 19 | 0.21277 | 0.25360 | **0.20257** | 0.21848 | 0.21022 | 0.21634 |
| | 25 | | | | | 0.21075 | |
| | 50 | | | | | 0.21179 | |
| | 74 | | | | | 0.21246 | |
| | 100 | | | | | 0.21234 | |
| | 1 | | | | | 0.31280 | |
| | 3 | | | | | 0.31558 | |
| | 6 | | | | | 0.31658 | |
| t + 20 min | 12 | 0.31534 | 0.33823 | 0.34178 | **0.31206** | 0.31745 | 0.32577 |
| | 25 | | | | | 0.31906 | |
| | 50 | | | | | 0.31990 | |
| | 100 | | | | | 0.32101 | |
| | **1** | | | | | **0.36711** | |
| | 3 | | | | | 0.36968 | |
| | 6 | | | | | 0.37245 | |
| t + 30 min | 12 | 0.38089 | 0.39786 | 0.37520 | 0.37064 | 0.37227 | 0.38499 |
| | 25 | | | | | 0.37367 | |
| | 50 | | | | | 0.37352 | |
| | 100 | | | | | 0.37538 | |
| | **1** | | | | | **0.50552** | |
| | 3 | | | | | 0.50730 | |
| | 6 | | | | | 0.51189 | |
| t + 60 min | 12 | 0.52320 | 0.53567 | 0.51731 | 0.51284 | 0.51289 | 0.52440 |
| | 25 | | | | | 0.51480 | |
| | 50 | | | | | 0.51571 | |
| | 100 | | | | | 0.51872 | |

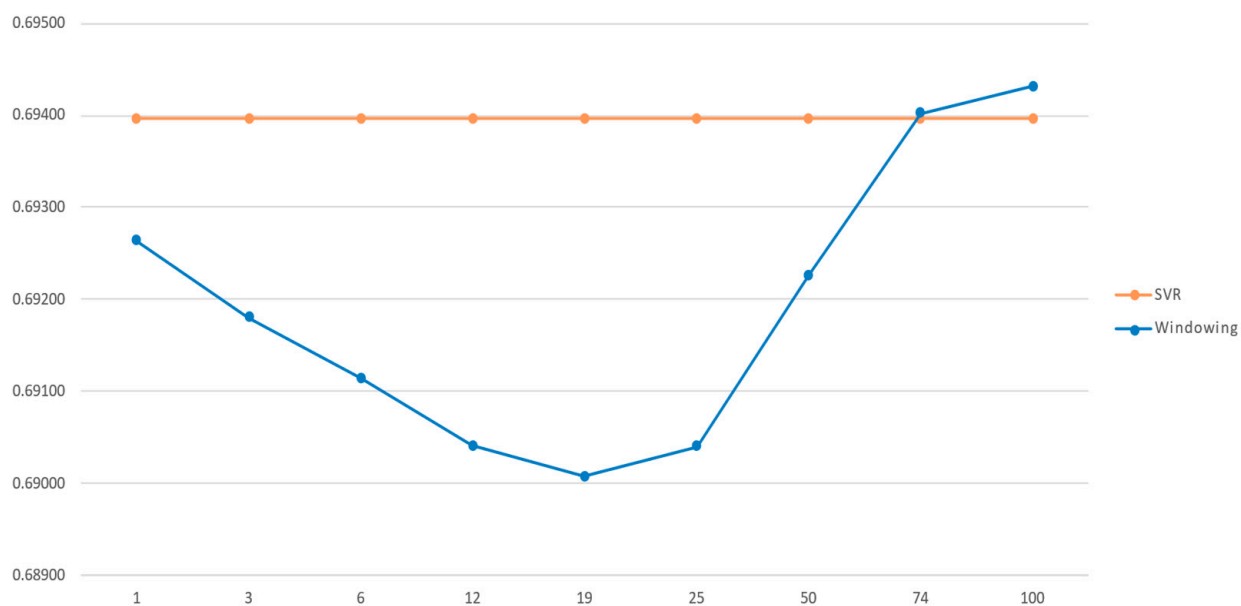

**Figure 8.** Parameter λ variance effect in method performance. SVR result is shown for reference.

### 4.2. Irradiance Predictions

During the search for best-performance methods, the optimized parameters of each of these methods needed to be known to allow the elaboration of the dynamic ensemble, which is built from merging the best-performance results at each instant and for each of the methods in question. The optimal parameters for each time horizon are shown in Table 8.

**Table 8.** Best parameters for each machine learning method.

| Method | Parameter | t + 10 | t + 20 | t + 30 | t + 60 |
|---|---|---|---|---|---|
| Random forest | best_max_depth | 5 | 5 | 5 | 5 |
| | best_n_estimators | 20 | 20 | 20 | 20 |
| KNN | best_n_neighbors | 37 | 37 | 49 | 48 |
| SVR | best_C | 0.1 | 0.1 | 0.1 | 0.1 |
| | best_epsilon | 0.1 | 0.1 | 0.1 | 0.1 |
| Elastic net | best_l1_ratio | 1 | 1 | 1 | 1 |

Efficiency evaluation for each of the solar irradiance forecasting methods were based on performance metrics for each time horizon under study (t + 10, t + 20, t + 30 and t + 60). Again, windowing proved to be the most efficient method for all time horizons, with the best method being found for the t + 10 time horizon, having the lowest RMSE value, using its parameterizations with λ = 50 initially. Then, fine-tuning was performed based on the variation of the windowing parameter to assess its influence on performance. The predominance of better performance for windowing in all time horizons and its comparisons can be seen in Table 9 and Figure 9.

**Table 9.** Comparison of RMSE (W/m$^2$) values, using different methods in different time horizons and windowing λ parameter variation. The best results for each time horizon are in bold.

| Time Horizon | λ | RF | KNN | SVR | Elastic Net | Windowing | Arbitrating |
|---|---|---|---|---|---|---|---|
| | **1** | | | | | **72.73186** | |
| | 3 | | | | | 72.93221 | |
| | 6 | | | | | 73.29363 | |
| t + 10 min | 12 | 75.02000 | 75.26000 | 74.19000 | 74.98000 | 73.21035 | 74.01000 |
| | 25 | | | | | 73.24620 | |
| | 50 | | | | | 73.48055 | |
| | 100 | | | | | 73.69330 | |
| | **1** | | | | | **80.07000** | |
| | 3 | | | | | 80.63000 | |
| | 6 | | | | | 81.19000 | |
| t + 20 min | 12 | 90.94000 | 83.50000 | 84.45000 | 84.53000 | 81.87000 | 83.19000 |
| | 25 | | | | | 82.56000 | |
| | 50 | | | | | 82.11000 | |
| | 100 | | | | | 82.57000 | |
| | **1** | | | | | **86.25000** | |
| | 3 | | | | | 87.00000 | |
| | 6 | | | | | 87.75000 | |
| t + 30 min | 12 | 90.15000 | 90.50000 | 91.49000 | 93.49000 | 88.33000 | 89.70000 |
| | 25 | | | | | 88.95000 | |
| | 50 | | | | | 88.70000 | |
| | 100 | | | | | 89.01000 | |
| | **1** | | | | | **105.51000** | |
| | 3 | | | | | 106.62000 | |
| | 6 | | | | | 107.76000 | |
| t + 60 min | 12 | 112.05000 | 112.13000 | 112.76000 | 118.08000 | 108.89000 | 111.13000 |
| | 25 | | | | | 109.32000 | |
| | 50 | | | | | 110.12000 | |
| | 100 | | | | | 110.30000 | |

Just like the evaluation employing RMSE, values of R$^2$, MAE, and MAPE were also analyzed. After the best performance was found for the windowing method, an in-depth analysis was performed based on the variation of its parameter λ to assess the influence on its internal performance. Since the time horizon that presented the best performance was t + 10, this was the focus of the analysis, as shown in Figures 10–13. The detailed data for all tested time horizons is shown in Tables 10–12.

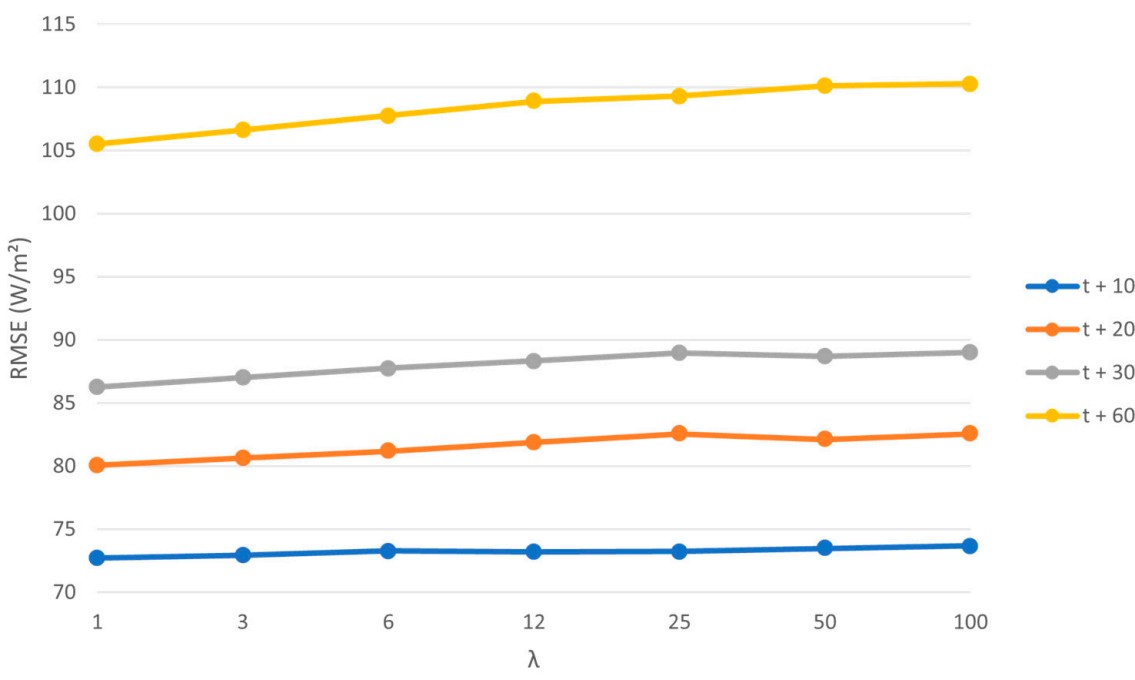

**Figure 9.** Windowing λ parameter variation influence in RMSE for all the studied time horizons in solar irradiation data analysis.

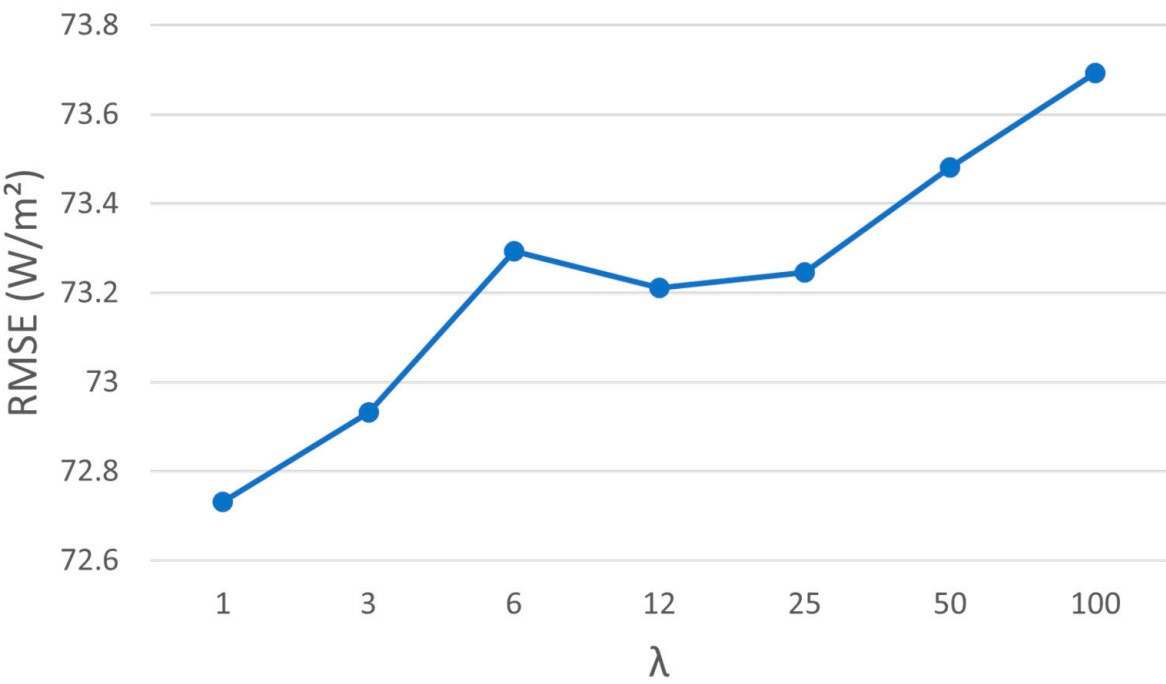

**Figure 10.** Windowing λ parameter influence in RMSE value in time horizon t + 10.

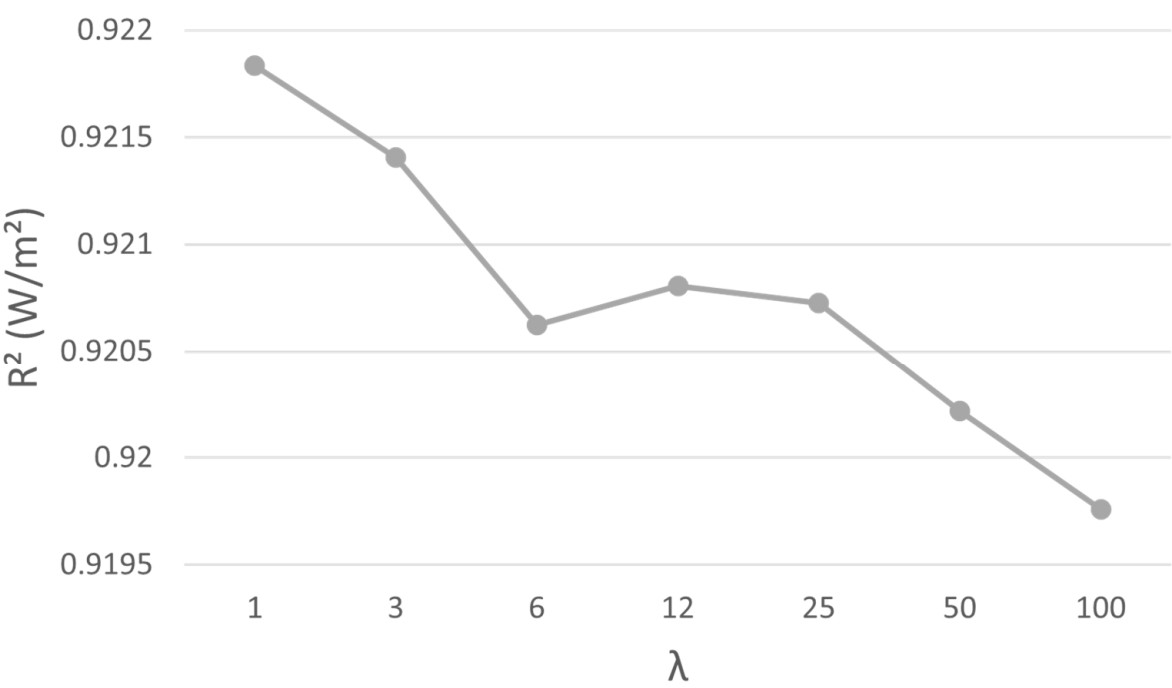

**Figure 11.** Windowing λ parameter influence in $R^2$ value in time horizon t + 10.

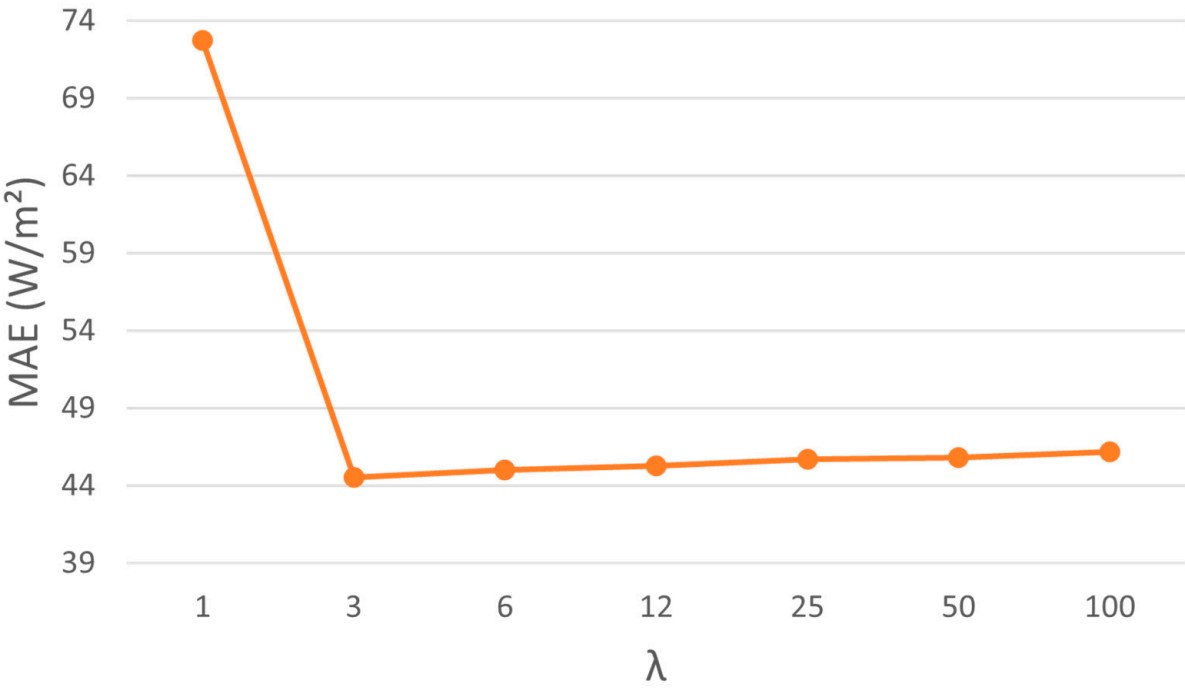

**Figure 12.** Windowing λ parameter influence in $R^2$ value in time horizon t + 10.

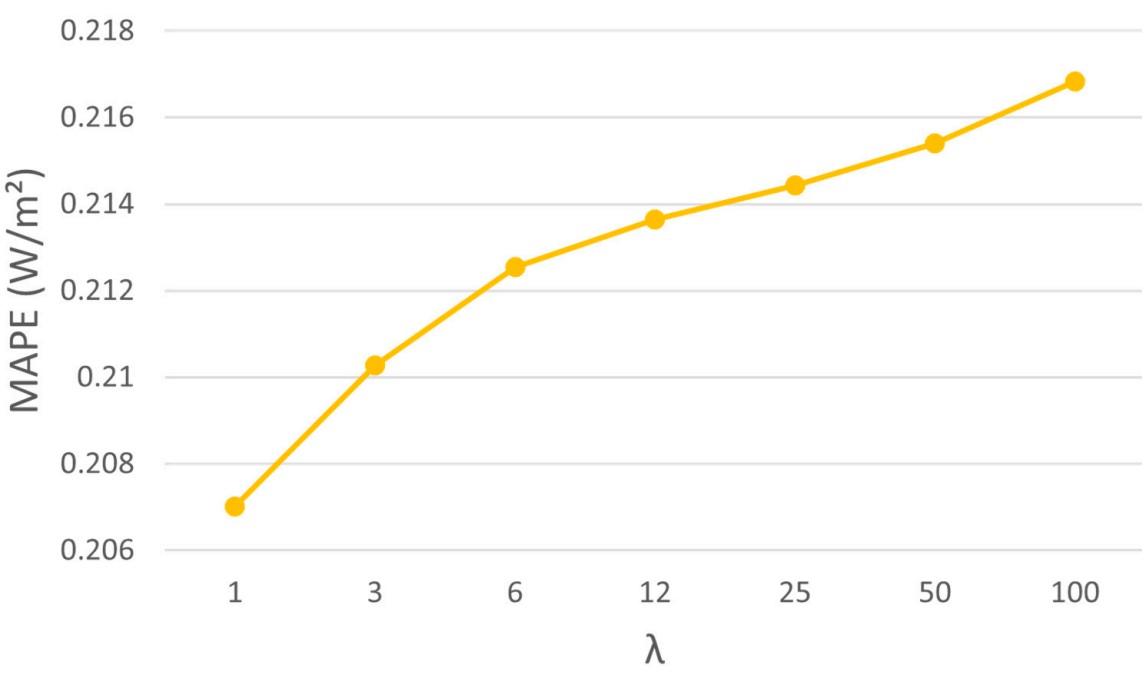

**Figure 13.** Windowing λ parameter influence in MAPE value in time horizon t + 10.

**Table 10.** Comparison of $R^2$ (W/m$^2$) values, using different methods in different time horizons and windowing λ parameter variation. The best results for each time horizon are in bold.

| Time Horizon | λ | RF | KNN | SVR | Elastic Net | Windowing | Arbitrating |
|---|---|---|---|---|---|---|---|
| | 1 | | | | | **0.92184** | |
| | 3 | | | | | 0.92141 | |
| | 6 | | | | | 0.92062 | |
| t + 10 min | 12 | 0.92000 | 0.92000 | 0.92000 | 0.92000 | 0.92080 | 0.92000 |
| | 25 | | | | | 0.92073 | |
| | 50 | | | | | 0.92022 | |
| | 100 | | | | | 0.91976 | |
| | 1 | | | | | **0.91000** | |
| | 3 | | | | | **0.91000** | |
| | 6 | | | | | 0.90000 | |
| t + 20 min | 12 | 0.88000 | 0.90000 | 0.90000 | 0.90000 | 0.90000 | 0.90000 |
| | 25 | | | | | 0.90000 | |
| | 50 | | | | | 0.90000 | |
| | 100 | | | | | 0.90000 | |
| | 1 | | | | | **0.89000** | |
| | 3 | | | | | **0.89000** | |
| | 6 | | | | | **0.89000** | |
| t + 30 min | 12 | 0.88000 | 0.88000 | 0.88000 | 0.87000 | **0.89000** | 0.88000 |
| | 25 | | | | | **0.89000** | |
| | 50 | | | | | 0.88000 | |
| | 100 | | | | | **0.89000** | |

**Table 10.** *Cont.*

| Time Horizon | λ | RF | KNN | SVR | Elastic Net | Windowing | Arbitrating |
|---|---|---|---|---|---|---|---|
| | **1** | | | | | **0.85000** | |
| | 3 | | | | | 0.84000 | |
| | 6 | | | | | 0.84000 | |
| t + 60 min | 12 | 0.83000 | 0.83000 | 0.82000 | 0.51223 | 0.84000 | 0.83000 |
| | 25 | | | | | 0.83000 | |
| | 50 | | | | | 0.83000 | |
| | 100 | | | | | 0.83000 | |

**Table 11.** Comparison of MAE (W/m$^2$) values, using different methods in different time horizons and windowing λ parameter variation. The best results for each time horizon are in bold.

| Time Horizon | λ | RF | KNN | SVR | Elastic Net | Windowing | Arbitrating |
|---|---|---|---|---|---|---|---|
| | 1 | | | | | 72.73186 | |
| | **3** | | | | | **44.52301** | |
| | 6 | | | | | 45.00717 | |
| t + 10 min | 12 | 48.29000 | 48.47000 | 44.16000 | 49.31000 | 45.27759 | 46.24000 |
| | 25 | | | | | 45.67924 | |
| | 50 | | | | | 45.79140 | |
| | 10 | | | | | 46.16632 | |
| | **1** | | | | | **52.53000** | |
| | 3 | | | | | 53.31000 | |
| | 6 | | | | | 54.12000 | |
| t + 20 min | 12 | 65.19000 | 55.63000 | 59.67000 | 58.86000 | 55.27000 | 56.20000 |
| | 25 | | | | | 56.88000 | |
| | 50 | | | | | 55.59000 | |
| | 10 | | | | | 56.79000 | |
| | **1** | | | | | **58.14000** | |
| | 3 | | | | | 59.02000 | |
| | 6 | | | | | 59.91000 | |
| t + 30 min | 12 | 62.09000 | 61.58000 | 64.77000 | 67.13000 | 60.85000 | 60.91000 |
| | 25 | | | | | 61.34000 | |
| | 50 | | | | | 61.84000 | |
| | 10 | | | | | 61.51000 | |
| | **1** | | | | | **74.59000** | |
| | 3 | | | | | 75.92000 | |
| | 6 | | | | | 77.11000 | |
| t + 60 min | 12 | 81.28000 | 79.84000 | 81.44000 | 89.07000 | 78.47000 | 79.80000 |
| | 25 | | | | | 79.08000 | |
| | 50 | | | | | 79.48000 | |
| | 10 | | | | | 79.63000 | |

**Table 12.** Comparison of MAPE (W/m$^2$) values, using different methods in different time horizons and windowing λ parameter variation. The best results for each time horizon are in bold.

| Time Horizon | λ | RF | KNN | SVR | Elastic Net | Windowing | Arbitrating |
|---|---|---|---|---|---|---|---|
| | 1 | | | | | **0.20701** | |
| | 3 | | | | | 0.21027 | |
| | 6 | | | | | 0.21254 | |
| t + 10 min | 12 | 0.22000 | 0.24000 | 0.21000 | 0.23000 | 0.21364 | 0.22000 |
| | 25 | | | | | 0.21444 | |
| | 50 | | | | | 0.21541 | |
| | 100 | | | | | 0.21684 | |
| | 1 | | | | | **0.25000** | |
| | 3 | | | | | **0.25000** | |
| | 6 | | | | | 0.26000 | |
| t + 20 min | 12 | 0.32000 | 0.28000 | 0.28000 | 0.27000 | 0.26000 | 0.27000 |
| | 25 | | | | | 0.27000 | |
| | 50 | | | | | 0.26000 | |
| | 100 | | | | | 0.27000 | |
| | 1 | | | | | **0.27000** | |
| | 3 | | | | | 0.28000 | |
| | 6 | | | | | 0.28000 | |
| t + 30 min | 12 | 0.29000 | 0.30000 | 0.29000 | 0.33000 | 0.28000 | 0.29000 |
| | 25 | | | | | 0.29000 | |
| | 50 | | | | | 0.29000 | |
| | 100 | | | | | 0.29000 | |
| | 1 | | | | | **0.32000** | |
| | 3 | | | | | **0.32000** | |
| | 6 | | | | | 0.33000 | |
| t + 60 min | 12 | 0.34000 | 0.35000 | 0.34000 | 0.54747 | 0.33000 | 0.34000 |
| | 25 | | | | | 0.34000 | |
| | 50 | | | | | 0.34000 | |
| | 100 | | | | | 0.34000 | |

　　　　Some authors applied elastic Nnet in time-varying combinations [16], using RMSE as a performance metric. They found that, for PV forecasts, it obtained 13.4% more precise forecasts than the simple average and for the wind forecast, it obtained 6.1% better forecasts.

　　　　In [21], an ensemble method which used MAPE as the comparative efficiency metric for wind speed data was studied with a value of 9.345%, and solar with 7.186%, which proved to be the most efficient.

　　　　In this study, performance improvements were obtained for the most efficient method (windowing) compared to the second most efficient for wind speed of 0.56% and, for solar irradiation, 1.86%.

*4.3. Comparison with Results from the Literature*

　　　　Performance of the windowing approach was compared with other wind forecasting models found in the literature. It is important to disclose that a direct comparison between different predictive models is not an easy task, since each applied approach has its own

objectives, hyperparameters, and input data [22]. To facilitate the comparison against the results found in the literature, Table 13 compiles the results previously presented for the proposed windowing model. The results found in literature for wind speed forecasting are compiled and presented in Table 14, where RMSE and MAE are in m/s.

**Table 13.** Compilation of the windowing's results for different time horizons.

| Metric | Time Horizon | Wind Speed | GHI |
|---|---|---|---|
| RMSE | t + 10 | 0.69007 m/s | 72.73186 W/m$^2$ |
| | t + 20 | 0.86817 m/s | 80.07 W/m$^2$ |
| | t + 30 | 0.97497 m/s | 86.25 W/m$^2$ |
| | t + 60 | 1.1515 m/s | 105.51 W/m$^2$ |
| R$^2$ | t + 10 | 0.84451 | 0.92184 |
| | t + 20 | 0.75388 | 0.91 |
| | t + 30 | 0.68958 | 0.89 |
| | t + 60 | 0.56685 | 0.85 |
| MAE | t + 10 | 0.51272 m/s | 44.52301 W/m$^2$ |
| | t + 20 | 0.64663 m/s | 52.53 W/m$^2$ |
| | t + 30 | 0.72594 m/s | 58.14 W/m$^2$ |
| | t + 60 | 0.86784 m/s | 74.59 W/m$^2$ |
| MAPE | t + 10 | 0.21022 | 0.20701 |
| | t + 20 | 0.3128 | 0.25 |
| | t + 30 | 0.36711 | 0.27 |
| | t + 60 | 0.50552 | 0.32 |

**Table 14.** Compilation of results for wind speed forecasting.

| Model | Metric Value | Author |
|---|---|---|
| GNN SAGE GAT | RMSE<br>0.638 for t + 60 forecasting horizon<br>MAE<br>0.458 for t + 60 forecasting horizon | Oliveira Santos et al. [22] |
| ED-HGNDO-BiLSTM | RMSE<br>0.696 average for t + 10 forecasting horizon<br>1.445 average for t + 60 forecasting horizon<br>MAE<br>0.717 average for t + 10 forecasting horizon<br>0.953 average for t + 60 forecasting horizon<br>MAPE<br>0.590 average for t + 10 forecasting horizon<br>9.769 average for t + 60 forecasting horizon | Neshat et al. [37] |
| Statistical model for wind speed forecasting | RMSE<br>1.090 for t + 60 forecasting horizon | Dowell et al. [38] |
| Hybrid wind speed forecasting model using area division (DAD) method and a deep learning neural network | RMSE<br>0.291 average for t + 10 forecasting horizon<br>0.355 average for t + 30 forecasting horizon<br>0.426 average for t + 60 forecasting horizon<br>MAE<br>0.221 average for t + 10 forecasting horizon<br>0.293 average for t + 30 forecasting horizon<br>0.364 average for t + 60 forecasting horizon | Liu et al. [39] |

**Table 14.** *Cont.*

| Model | Metric Value | Author |
|---|---|---|
| Hybrid model CNN-LSTM | RMSE<br>0.547 for t + 10 forecasting horizon<br>0.802 for t + 20 forecasting horizon<br>0.895 for t + 30 forecasting horizon<br>1.114 for t + 60 forecasting horizon<br>MAPE<br>4.385 for t + 10 forecasting horizon<br>6.023 for t + 20 forecasting horizon<br>7.510 for t + 30 forecasting horizon<br>11.127 for t + 60 forecasting horizon | Zhu et al. [40] |

Analyzing the results for reference [22], in which wind speed was forecasted in the Netherlands using an ensemble approach merging graph theory and attention-based deep learning, we can observe that the proposed windowing ensemble model is not able to surpass the results for neither RMSE nor MAE for t + 60 forecasting horizon. The accentuated difference between these two models can be explained because the GNN SAGE GAT model, being developed to handle graph-like data structure, excels in retrieving complex spatiotemporal relationships underlaying the dataset, drastically improving its forecasting capacity when compared with other ML and DL models alike.

In reference [37], the authors proposed a wind forecasting for a location in Sweden, with a model based on a bi-directional recurrent neural network, a hierarchical decomposition technique, and an optimization algorithm. When compared with their results, the windowing model proposed in this paper offers improvement over the reference results for t + 10 forecasting horizon by 1% and by 20% for t + 60. When MAE and MAPE are analyzed, the windowing indicates improvement over these metrics for t + 10 and t + 60, increasing by 28% the MAE value for t + 10, and 9% for t + 60. Regarding MAPE, the improvement is 64% for t + 10 and 95% for t + 60.

In the work of Liu et al. [39], another deep learning-based predictive model was proposed. It used a hybrid approach composed of data area division to extract historical wind speed information and an LSTM layer optimized via a genetic algorithm to process the temporal aspect of the dataset to forecast wind speed in Japan. Compared to this reference, the windowing model showed no improvement for wind speed forecasting. However, the windowing approach offers competitive forecasting for the assessed time windows, being in the same order of magnitude as the ones in the reference. In work [40], the authors proposed the employment of another hybrid forecasting architecture composed of CNN and LSTM deep learning models for wind speed estimation in the USA. Their results, when compared against the windowing methodology, are very similar for all forecasting horizons, showing that both windowing and CNN–LSTM offer good results for wind speed estimation for these time intervals.

In Dowell et al. [38], a statistical model for estimation of future wind speed values in the Netherlands was proposed. For the available t + 60 time horizon, we observe that, again, the forecasted wind speeds for the reference and proposed windowing models are very similar, suggesting both models as valuable tools for wind speed forecasting.

For GHI forecasting, the results found in the literature are presented in Table 15.

**Table 15.** Compilation of results for GHI forecasting.

| Model | Metric Value | Author |
|---|---|---|
| CNN-1D | RMSE ($R^2$)<br>36.24 (0.98) for t + 10 forecasting horizon<br>39.00 (0.98) for t + 20 forecasting horizon<br>38.46 (0.98) for t + 30 forecasting horizon | Marinho et al. [23] |
| MEMD-PCA-GRU | RMSE ($R^2$)<br>31.92 (0.99) for t + 60 forecasting horizon | Gupta and Singh [41] |
| Physical-based forecasting model | RMSE<br>75.91 for t + 30 forecasting horizon<br>89.81 for t + 60 forecasting horizon<br>MAE<br>48.85 for t + 30 forecasting horizon<br>57.01 for t + 60 forecasting horizon | Yang et al. [42] |
| Physical-based forecasting model | RMSE<br>114.06 for t + 60 forecasting horizon | Kallio-Meyers et al. [43] |
| Deep learning transformer-based forecasting model | MAE<br>34.21 for t + 10 forecasting horizon<br>43.64 for t + 20 forecasting horizon<br>49.53 for t + 30 forecasting horizon | Liu et al. [44] |

In work [23], a deep learning standalone model of CNN was applied to estimate future GHI values in the USA. Comparing the GHI forecasting results achieved via windowing with this reference, we observe that the proposed model was not able to provide superior forecasting performance. However, the windowing results are still competitive since both approaches were able to reach elevated coefficient of determination values for all the assessed forecasting horizons, with a slight advantage for the deep learning model.

In reference [41], the authors combined principal component analysis (PCA) with multivariate empirical model decomposition (MEMD) and gated recurrent unit (GRU) to predict GHI in India. In their methodology, the PCA extracted the most relevant features from the dataset after it was filtered via the MEMD algorithm. Lastly, the future irradiance was estimated via the deep learning model of GRU. Compared to their approach, the windowing model could not improve the GHI forecasting within a t + 60 time window. Also, the reference model MEMD-PCA-GRU provided an elevated $R^2$ value of 99%, showing clearly superior performance over the proposed ensemble model.

When our model is compared with the physical-based forecasting models proposed in [42,43], we can conclude that windowing can achieve similar results for time horizons of t + 30 and t + 60. In [42], authors used the FY-4A-Heliosat method for satellite imagery to estimate GHI in China. Although the windowing model could not improve on GHI forecasting for t + 30 and t + 60 time windows, the proposed model was able to return relevant results for irradiance estimation in both cases. The second physical-based model proposed in [43] was applied to estimate GHI in Finland. In their methodology, the Heliosat method is again employed, together with geostationary weather data from satellite images. Compared to their proposed approach, the windowing model can improve GHI forecasting for t + 60 in 8%, providing significant advance in the irradiance estimation.

In work [44], the authors used the state-of-the-art transformer deep learning architecture together with sky images [45] for GHI estimation in the USA. Analyzing their results and the ones provided by the windowing method, we observe that the transformer-based model reaches the best GHI forecasting values for RMSE in all the assessed time windows.

After the comparison of the ensemble windowing approach with reference models found in the literature, we see that wind speed prediction is often competitive and usually improves wind speed prediction for the assessed forecasting horizons. The results for wind speed prediction using the ensemble model corroborate the results found in the literature, where the ensemble approach often reaches state-of-the-art forecasting in time-

series prediction applications [21,46–48]. Their improved performance comes from the combination of weaker predictive models to improve their overall forecasting capacity, also reducing the ensembled model's variance [49,50].

However, the proposed dynamic ensembled approach faced increased difficulty when determining future GHI values. This may be an indication that irradiance forecasting is a more complex non-linear natural phenomenon, requiring improved extraction of spatiotemporal information from the dataset. Since the proposed ensemble model does not have a deep learning model in its architecture it cannot properly identify and extract spatiotemporal information underlying the dataset, thus failing in providing better irradiance estimation. Deep learning models can often excel in this type of task, as proved in the results from Table 15. Extensive literature can be found regarding improvements of time-series forecasting problems when complex and deep approaches are employed [22,23,51,52].

## 5. Conclusions

This work proposed to evaluate the performance of two machine learning (ML) dynamic ensemble methods, using wind speed and solar irradiance data separately as inputs. Initially, wind speed and solar irradiance data from the same meteorological station were collected, the time horizons to be studied were determined (t + 10 min, t + 20 min, t + 30 min and t + 60 min), and then a recursive approach of lagged average values was applied to evaluate the models' predictors.

ML methods well known in other energy forecasting research works regarding wind and irradiance data were selected to compare their efficiency with two other methods that use a dynamic ensemble approach (windowing and arbitrating). The programming code in Python was developed to catalog the optimal efficiency parameters of each previously known model, based on error metrics and coefficient of determination. The dynamic ensemble methods (windowing and arbitrating), based on the optimal parameters of each previously calibrated models (random forest, k-nearest neighbors, support vector regression, and elastic net), generated a single model with greater efficiency for both wind and solar irradiance data.

For forecasting wind speed data, the most efficient method was found to be windowing for all time horizons, when evaluated by the criterion of the lowest RMSE value, and specifically for the time horizon t + 10, as evidenced in Figure 3. The greatest efficiency was found in an interval of 1 to 74 for the $\lambda$ parameter, reaching maximum performance for the value $\lambda = 19$, as seen in Figure 8, which suggests that the windowing parameterization directly influences the method's performance.

Structurally, solar radiation data is different from wind data, since they have cycles in nature and are different physical phenomena, presenting different correlations with their historical values, which impacts different trends for the $\lambda$ parameter in each of the variables.

For solar irradiation forecasting, the most efficient method was also windowing and the t + 10 min time horizon reached the lowest RMSE value. Unlike what was found for wind speed data, a greater linearity in the trend was perceived from the $\lambda$ windowing parameter variation plot when analyzing its RMSE values. Looking at the $\lambda$ interval under study, the best performance value (using RMSE criteria) of $\lambda = 1$ was found, as can be seen in Figure 10. Unlike all other plots, in Figure 12, there is a sudden jump between $\lambda$ from 1 to 3. Although the reference metric is RMSE, for some other metrics the use of $\lambda = 1$ may mean insufficient information for the model, since it will have as input variable just one previous time step (window size).

Using wind speed data, the efficiency gain of the most efficient model (windowing for the time horizon t + 10 min and $\lambda = 19$, see Table 4), when compared to the second highest efficiency (SVR), was 0.56% when using the lowest value RMSE metric. A similar trend could be observed for the model using solar irradiance data. The efficiency increase, comparing the most efficient model (windowing for the time horizon t + 10 min and

$\lambda = 1$, see Table 9) to the second highest efficiency (arbitrating), was about 1.72%, and when compared to the third most efficient method (SVR), it was about 1.96%.

Also, extensive comparisons with spatiotemporal models found in the literature show that the dynamic ensemble model for wind speed often provides superior forecasting performance for the assessed time horizons, deeming the proposed approach as a valuable tool for wind speed estimation. Regarding irradiance forecasting, the dynamic ensemble architecture proposed in this study could not surpass the deep learning-based models, which showed superior spatiotemporal identification, and consequently better estimated GHI values. However, the proposed windowing approach can provide competitive results and superior GHI forecasting when compared to physics-based predictive models.

For future works, the dynamic ensemble architecture can be improved with the addition of more complex machine learning models, such as deep learning and graph-based approaches, such as the one in works [22,51,52]. This may boost the windowing forecasting capacity for GHI and wind speed estimation once it is able to benefit from spatiotemporal data information underlying the dataset. The models were developed to treat the database in a generalized way. Specific studies with delimitation of seasons and/or times of day can be carried out as future studies. The development of an ensemble model able to provide accurate and precise estimations can then be employed in the development of real-time forecasting applications, helping the evaluation of wind and solar farms operation.

**Author Contributions:** Conceptualization, F.D.V.B., F.P.M. and P.A.C.R.; data curation, F.D.V.B. and F.P.M.; formal analysis, P.A.C.R.; methodology, F.D.V.B., F.P.M. and P.A.C.R.; software, F.D.V.B. and F.P.M.; supervision, P.A.C.R.; validation, P.A.C.R., J.V.G.T. and B.G.; visualization, P.A.C.R., J.V.G.T. and B.G.; writing—original draft, F.D.V.B., F.P.M. and V.O.S.; writing—review and editing, F.D.V.B., F.P.M., P.A.C.R., V.O.S., J.V.G.T. and B.G.; project administration, P.A.C.R.; funding acquisition, P.A.C.R., B.G. and J.V.G.T. All authors have read and agreed to the published version of the manuscript.

**Funding:** This research was funded by the Natural Sciences and Engineering Research Council of Canada (NSERC) Alliance, grant No. 401643, in association with Lakes Environmental Software Inc., by the Coordenação de Aperfeiçoamento de Pessoal de Nível Superior—Brasil (CAPES)—Finance Code (Grant No. 001), and by the Conselho Nacional de Desenvolvimento Científico e Tecnológico—Brasil (CNPq), grant no. 303585/2022-6.

**Institutional Review Board Statement:** Not applicable.

**Informed Consent Statement:** Not applicable.

**Data Availability Statement:** The data of wind speed and irradiation from Petrolina—PE—Brazil are downloaded from SONDA (National Organization of Environmental Data System)\\portal (http://sonda.ccst.inpe.br/, accessed on 12 July 2023).

**Conflicts of Interest:** The authors declare no conflict of interest.

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
