# Peer review of "Machine Learning Dynamic Ensemble Methods for Solar Irradiance and Wind Speed Predictions"

_atmosphere, doi:10.3390/atmos14111635_

Round 1

Reviewer 1 Report

Comments and Suggestions for Authors

This paper is interesting and presents novelty, precisely concerning the evaluation of the performance of dynamic ensemble approach. However before, accepting in Atmosphere, the authors must considered the following suggestions to revise the paper:

Introduction part

line 36 Replace by : A very undesired effect when it comes to electrical generation from alternative resources, such wind speed and global solar radiation,  is the impact of intermittency generation [1]...

[1] R. Calif, F.G. Schmitt and O. Duran Medina, -5/3 Kolmogorov turbulent behavior and Intermittent Sustainable Energies, in book Sustainable Energy-Technological Issues, Applications and Case Studies, Editors: Ahmed Zobaa, Shady Abdel Aleem, Sara N. Affi, Intech, 2016 DOI: 10.5772/65755

Location and data part

Please precise in wind speed and solar data, the time sampling and the duration of data collection corresponding. Moreover, it would be interesting to display an extract of wind and global solar radiation sequences.

Methodology part

Recall briefly the theoretical framework of forecasting models used.

Reviewer 2 Report

Comments and Suggestions for Authors

Comments and suggestions are in attached file. 

Comments on the Quality of English Language

Comments and suggestions are in attached file. 

Reviewer 3 Report

Comments and Suggestions for Authors

Authors have presented an interesting manuscript entitled “Machine Learning Dynamic Ensemble Methods for Solar Irradiance and Wind Speed Predictions”. The study focuses on the machine learning algorithms applied for solar irradiance and wind speed prediction. We think that the study has been performed at high scientific level. However, we have a number of notices:

1. First of all, we have questions about the initial measurement data used for machine learning. The measurement itself is very important. Please pay more attention to the description of the initial measurement data. Firstly, it would be useful to provide an image of the equipment located at the site. Secondly, please clarify how the initial measurement accuracy of wind speed affects the ability to approximate the characteristics under study using neural networks. Please indicate what sensors do you used?

2. In the text authors used excessive rounding (five decimal places), for example, for MAPE, RMSE. Also please verify units for used statistical characteristics (RMSE, MAPE...) .

3. What physical reasons are associated with the fact that MAE and RMSE parameters have minimum at λ = 19. What do the values λ = 1, λ = 19 , λ = 100 correspond to? Is it possible to perform calculations for λ = 1000?

4. Figure 8 demonstrate parameter λ variance effect in method performance. Authors presented also results for support vector regression (SVR) method. Why does the SVR method have no dependency on parameter λ?

5. Have you compared your results with calculations performed using multilayer neural networks (Multilayered perceptron)?

6. Also, we recommend authors to expand Introduction. We understand that there are a lot of papers devoted to the machine learning methods for diagnostic and prediction of atmospheric parameters. But we recommend to look at the following papers (it is possible that the papers may be useful): DOI 10.1088/1538-3873/acb384,https://doi.org/10.1029/2023MS003606).

We recommended the manuscript for publication after corrections.

Round 2

Reviewer 3 Report

Comments and Suggestions for Authors

Authors have made corrections and provided  explanations. I propose to accept the manuscript for publication.